# Garlic-Induced Enhancement of *Bifidobacterium*: Enterotype-Specific Modulation of Gut Microbiota and Probiotic Populations

**DOI:** 10.3390/microorganisms12101971

**Published:** 2024-09-28

**Authors:** Jina Ha, Jinwoo Kim, Seongok Kim, Kwang Jun Lee, Hakdong Shin

**Affiliations:** 1Department of Food Science and Biotechnology, College of Life Science, Sejong University, Seoul 05006, Republic of Korea; 2Carbohydrate Bioproduct Research Center, College of Life Science, Sejong University, Seoul 05006, Republic of Korea; 3Division of Zoonotic and Vector Borne Diseases Research, Center for Infectious Diseases Research, National Institute of Health, Cheongju 28159, Republic of Korea

**Keywords:** gut microbiome, enterotype, probiotics, prebiotics, garlic, *Bifidobacterium*

## Abstract

The gut microbiome is a dynamic ecosystem crucial for maintaining its host’s health by regulating various immune and metabolic functions. Since diet plays a fundamental role in shaping the gut microbiome, understanding the relationship between food consumption and microbiome structure is essential. Although medicinal plants are widely recognized for their broad health benefits, their specific impact on the gut microbiome remains unclear. In this study, we investigated the effects of garlic (*Allium sativum*) on the gut microbiome using an in vitro human fecal incubation model. Our findings revealed that the impact of garlic on gut microbial structure varied depending on the dominant gut microbiome components (enterotypes). The *Bacteroides*-dominant enterotype exhibited significant changes in overall microbial diversity in response to garlic, while the *Prevotella*-dominant enterotype remained unaffected. Additionally, the garlic treatment led to specific alterations in microbiota composition, such as an increase in beneficial probiotics like *Bifidobacterium*. We validated garlic’s prebiotic potential by promoting the growth of *Bifidobacterium adolescentis* under in vitro culture conditions. Our study highlights the importance of understanding enterotype-specific responses to diet and suggests that garlic may serve as a dietary supplement for modulating gut microbiota and promoting the growth of beneficial probiotics.

## 1. Introduction

The gut microbiome is an ecological community comprising microbes, their genetic material, and the surrounding environment [1]. It is considered a key factor in maintaining its host’s fitness, supporting a range of physiological functions, such as nutrition, metabolism, immune regulation, and even the nervous system [2,3,4]. Since disruptions in this balanced system, known as dysbiosis, are associated with numerous health issues, maintaining a healthy microbiome is important for overall well-being and disease prevention [5]. One of the most significant factors shaping the composition and diversity of gut microbial communities is diet [6,7]. The nutrients provided by dietary intake serve as substrates that support the growth of specific microbial species, thereby influencing the overall microbial landscape. Given the profound impact of diet on the gut microbiome, understanding the specific effects of various dietary components is essential for developing targeted nutritional strategies that could enhance health outcomes and reduce the risk of diseases associated with dysbiosis.

The composition of the gut microbiota varies significantly among individuals, which poses a challenge for studying the relationship between microbiome and external factors, such as diet [8]. To address this complexity, the concept of enterotypes was introduced, classifying human gut microbiomes based on the predominance of specific bacterial genera [9]. The three primary enterotypes are *Bacteroides*-dominant type (enterotype 1), *Prevotella*-dominant type (enterotype 2), and *Ruminococcus*-dominant type (enterotype 3) [9,10]. This enterotype-based approach clusters individuals with similar microbiome compositions, thereby reducing individual variability and enabling more precise analysis [11]. By stratifying populations into these distinct enterotypes, we can more effectively explore the complex relationships between diet, microbiome composition, and health outcomes [12,13]. This approach not only uncovers patterns that might be obscured by individual variability but also supports the development of targeted dietary and therapeutic interventions.

Garlic (*Allium sativum*) is a medicinal plant with a long history of worldwide use, valued both as a food ingredient and for its therapeutic properties. It is a natural source of bioactive compounds, including sulfur-based metabolites such as allicin, alliin, diallyl sulfide, and diallyl trisulfide [14]. These compounds provide garlic with a range of therapeutic benefits, including antimicrobial, antibacterial, antioxidant, anti-inflammatory, and anticancer properties [15,16]. With an annual production of approximately 28 million tons, garlic is widely consumed around the world due to its extensive availability [17]. While the beneficial effects of garlic on human health are well known, its role in shaping the composition and function of the gut microbiome is not fully understood. There is limited insight into how garlic may serve as a prebiotic, i.e., selectively promoting the growth of beneficial microbial populations and, consequently, influencing overall human health. Given garlic’s extensive use and functionality, exploring its specific interactions with the microbiome could provide valuable insights into its health benefits.

In this study, we investigated the impact of garlic on the human gut microbiome using an in vitro fecal incubation model. We analyzed the effects of the garlic extract treatment on the gut microbiome based on the fecal enterotype. Additionally, we validated the prebiotic effects of garlic on specific *Bifidobacterium* species. Our findings suggest that garlic extract can modulate the gut microbiome, potentially contributing to improved gut health and disease prevention.

## 2. Materials and Methods

### 2.1. Study Population

This study involved 48 healthy Korean participants, including 25 males and 23 females, aged between 23 and 32 years. All participants had no history of gastrointestinal problems and had not taken antibiotics for one month prior to the experiment. Fecal samples were collected using sterile cotton swabs and immediately frozen at −80 °C for further analysis. The present study was approved by the Institutional Review Board (IRB) of Sejong University (IRB no. SJU-BR-E-2020-025).

### 2.2. In Vitro Fecal Incubation

For in vitro fecal incubation, samples were transferred into an anaerobic workstation and prepared as described previously [18]. In summary, fresh stools were homogenized in a MiPro medium and filtered through a sterile nylon mesh cloth (985 µm). The homogenized samples were then inoculated into 96-deep-well plates at a final fecal concentration of 2.0% (*w*/*v*). When necessary, commercial garlic powder was added at final concentrations of 1 mg/mL. The plates were incubated anaerobically at 37 °C and the samples were taken at 0 and 24 h.

### 2.3. 16S rRNA Amplicon Analysis

The total genomic DNA was extracted using the DNeasy PowerSoil HTP 96 kit (Qiagen, Hilden, Germany) following the manufacturer’s instructions. The V4 hypervariable region of the 16S rRNA gene was amplified using 515F/806R primers, and the resulting amplicons were sequenced on the Illumina MiSeq platform (2 × 300 cycles, paired-end) according to Earth Microbiome Project protocols [19]. The 16S rRNA sequences were analyzed using the QIIME 2 (v2020.6) software package [20]. Raw reads were demultiplexed and quality-filtered using the q2-demux plugin, followed by trimming and denoising with DADA2 [21]. The amplicon sequence variants (ASVs) were aligned with MAFFT, and each ASV was taxonomically assigned using the SILVA 132 database [22,23]. Sequences were rarefied to a depth of 3521 per sample to ensure fair comparisons. Further microbiome data analysis was performed using MicrobiomeAnalyst 2.0 for comprehensive statistical, functional, and meta-analysis of the microbiome data [24].

### 2.4. Bacterial Growth Curve Assay

The bacterial strains used in this study are listed in Appendix A. The strains were cultured anaerobically at 37 °C in MRS broth, which was supplemented with 0.1% (*w*/*v*) L-cysteine but excluded glucose. When necessary, the media was supplemented with garlic powder dissolved in water at the indicated concentration. If not otherwise specified, commercial garlic powder was added at a final concentration of 1 mg/mL. Bacterial growth was monitored by measuring the absorbance rate at a wavelength of 580 nm.

### 2.5. Quantification and Statistical Analysis

Statistical analysis was performed using Python (version 3.9.12) with the following libraries: numpy (version 1.21.5), pandas (version 1.4.2), scipy (version 1.7.3), and statsmodels (version 0.13.2). The Shapiro–Wilk test was utilized to assess the normality of data distributions. Comparisons between the two groups were performed using either Student’s *t*-test or Welch’s *t*-test, depending on the homogeneity of variances as determined by Levene’s test. For nonparametric data, pairwise comparisons were conducted using the Kruskal–Wallis test. All *p*-values were adjusted using the Benjamini–Hochberg procedure to control the false discovery rate.

## 3. Results

### 3.1. Stratification of Gut Microbiome Enterotypes 

This study involved 48 healthy Korean participants, including 25 males and 23 females, aged between 23 and 32 years. To minimize individual variability, we stratified the participants’ gut microbiomes into enterotypes based on the composition and abundance of their microbial communities. The samples were clustered into two distinct groups based on their relative abundance at the genus level, using the Jensen–Shannon divergence (JSD) distance and the partitioning around medoids (PAM) clustering algorithm (Figure 1A and Appendix A). After clustering the samples, we conducted a detailed analysis of the microbial composition within each group to identify the key microbial signatures that distinguish them (Figure 1B). We observed that the levels of the predominant genera, specifically *Bacteroides* and *Prevotella*, differed significantly between the two groups (Figure 1C,D). Of the participants, 40 samples had a *Bacteroides*-dominant enterotype (B-type), while 8 samples had a *Prevotella*-dominant enterotype (P-type).

We further analyzed the relationship between the clustered groups and the *Prevotella* ratio (the ratio of *Prevotella* levels to the sum of *Bacteroides* and *Prevotella* levels) to validate the distinctiveness of the clusters (Figure 1E). Although the *Prevotella* ratio was distinguished between the two groups, two samples (S.39 and S.40) fell outside the normal range. Sample S.39, clustered as *Bacteroides*-dominant enterotype (B-type), had a high *Prevotella* ratio of 0.20, compared to the 0 to 0.07 range observed in others of the same group. Sample S.040, classified as *Prevotella*-dominant enterotype (P-type), exhibited a lower *Prevotella* ratio of 0.4, whereas others in this enterotype ranged from 1.12 to 8.88. Moreover, these two samples were outliers in the principal coordinate analysis (PCoA) plot, which was consistent with the *Prevotella* ratio results (Figure 1A). Consequently, we excluded these subjects to ensure a more accurate assessment of garlic’s effect on the gut microbiota based on enterotype.

### 3.2. Enterotype-Specific Changes in Gut Microbiome Alpha Diversity with Garlic Treatment

We incubated fecal samples with garlic using the in vitro incubation model to determine its impact on the intestinal microbiota. We calculated alpha diversity to assess the richness and phylogenetic diversity of the intestinal microbiome according to the enterotypes. In the B-type, garlic treatment significantly reduced microbial richness, as indicated by lower species counts and indices compared to the control group (Figure 2A). Additionally, phylogenetic diversity, measured by Faith’s PD test, showed significant differences (Figure 2B). However, in the P-type, garlic treatment did not significantly affect alpha diversity.

### 3.3. Enterotype-Specific Effects of Garlic on Gut Microbiota Beta Diversity 

We conducted a beta diversity analysis to identify the effects of garlic treatment on microbial community compositions. Although the 2D principal coordinates analysis (PCoA) plot did not clearly separate the groups, pairwise permutational multivariate analysis of variance (PERMANOVA) revealed significant differences in microbial community composition within the *Bacteroides*-dominant enterotype (B-type). Unweighted UniFrac analysis, which focused on the presence or absence of microbial taxa, showed significant differences between the garlic-treated group and the control group (Figure 2C). Weighted UniFrac analysis, which considered both the presence and abundance of taxa, also exhibited significant differences (Figure 2D). In contrast, the P-type showed no significant differences in either the weighted or unweighted analyses. Furthermore, we examined the beta diversity using the Bray–Curtis index at the genus level. Garlic treatment had a substantial impact on the microbiome of the B-type only, with no significant effect observed in the P-type (Appendix A). These findings suggested that the impact of garlic on the intestinal microbiota may be influenced by the specific enterotype.

### 3.4. Microbial Composition Differential at the Phylum Levels

We performed a detailed analysis of the relative bacterial abundance and taxa at the phylum level (Figure 3A). Amplicon sequence variants (ASVs) identified seven distinct phyla, with Bacteroidota and Firmicutes being the dominant phyla. In the B-type enterotype, the combined relative abundance of Bacteroidota and Firmicutes accounted for 82.9% in the control group and 76.3% in the garlic-treated group. For the P-type enterotype, these phyla represented 71.3% in the control group and 69.2% in the garlic-treated group. Given the importance of the Firmicutes/Bacteroidota (F/B) ratio as a key marker of gut homeostasis [25], we assessed the impact of garlic treatment on this ratio (Figure 3B). Our analysis revealed that garlic treatment significantly affected the F/B ratio, but this effect was exclusive to the B-type enterotype.

Further investigation of the gut microbiome at the phylum level revealed significant changes in Actinobacteriota, Bacteroidota, and Desulfobacterota following garlic treatment (FDR < 0.05). Specifically, garlic treatment resulted in a 3.7-fold increase in the relative abundance of Actinobacteriota within the B-type enterotype compared to the control group (Figure 3C), while the relative abundances of Bacteroidota and Desulfobacterota decreased (Figure 3D,E). In contrast, no statistically significant changes in phyla were observed between the control and garlic-treated groups in the P-type enterotype.

### 3.5. Effects of Garlic Supplementation on Gut Microbiome Composition

We examined the relative taxa abundance at the genus level (Figure 4A) and analyzed key genus features using Linear Discriminant Analysis Effect Size (LEfSe) [26]. Applying a Linear Discriminant Analysis (LDA) score threshold of >3.0 and *p*-values < 0.05, we identified significant changes in key genera. In the P-type enterotype, garlic treatment resulted in notable changes in only two genera: *Bifidobacterium* and *Ruminococcus_torques_group* (Figure 4B). The B-type enterotype showed substantial alterations across 16 genera, including the aforementioned two genera (Figure 4C). Garlic treatment resulted in an increase in *Bifidobacterium* and a decrease in *Ruminococcus*. The increase in the relative abundance of *Bifidobacterium* within the B-type enterotype was consistent with phylum-level observations, where the Acidobacteria phylum, which included *Bifidobacterium*, showed an increase (Figure 3C). The similar patterns of change in *Bifidobacterium* and *Ruminococcus* across all enterotypes suggest that these genera might be the key benefits of garlic’s effects.

### 3.6. Garlic’s Selective Prebiotic Effects on Bifidobacterium Species

While garlic can exert its effects on various genera, the abundance of *Bifidobacterium* changed in response to garlic treatment across all enterotypes (Figure 3D). Given the well-known benefits of *Bifidobacterium* [27], we focused on exploring the relationship between its growth and garlic supplementation. We analyzed the microbiome amplicon sequence variants (ASVs) to identify features mapping for *Bifidobacterium* species. Twelve distinct features were confirmed at the species level using the Basic Local Alignment Search Tool (BLAST), identifying species such as *B. adolescentis*, *B. faecale*, *B. animalis*, *B. longum*, and *B. pseudocatenulatum* (Appendix A and Figure 5A–E). Comparing the relative abundance of these species with and without garlic incubation revealed that *B. adolescentis* (or *B. faecale*) and *B. pseudocatenulatum* were significantly affected by garlic treatment (Figure 5A,E). To validate the prebiotic effects of garlic on *Bifidobacterium*, we monitored the growth rates of several *Bifidobacterium* species with and without garlic treatment (Figure 5F–J). The results showed that garlic specifically promoted the growth of *B. adolescentis* (Figure 5I). These findings suggest that garlic can function as a prebiotic, particularly benefiting *B. adolescentis*, and may be useful in developing symbiotic formulations.

## 4. Discussion

In this study involving 48 healthy Korean participants, we classified their gut microbiomes into two distinct enterotypes: *Bacteroides*-dominant (B-type) and *Prevotella*-dominant (P-type) (Figure 1). We found that garlic treatment had a pronounced effect on the B-type enterotype, significantly reducing microbial richness and altering community composition, as evidenced by changes in alpha and beta diversity. In contrast, the P-type enterotype showed no significant response to garlic (Figure 2). At the phylum level, garlic treatment notably influenced the Firmicutes/Bacteroidota (F/B) ratio in the B-type enterotypes, while no significant changes were observed in the P-type enterotypes (Figure 3). Further genus-level analysis revealed an increase in *Bifidobacterium* across both enterotypes, with *B. adolescentis* showing significant growth promotion (Figure 4 and Figure 5). These results suggest that garlic could act as a prebiotic, especially promoting the growth of *Bifidobacterium*, and highlight its potential in developing symbiotic formulations.

Enterotypes, the classification of human gut microbiota into distinct compositional profiles, could provide significant insights into the relationship between microbiota composition and host factors. The three primary enterotypes are *Bacteroides*-dominant type (enterotype 1), *Prevotella*-dominant type (enterotype 2), and *Ruminococcus*-dominant type (enterotype 3) [9,10]. These enterotypes are not randomly distributed but are profoundly influenced by dietary patterns, which reflect geographic and cultural differences. For instance, a *Bacteroides*-dominated enterotype is prevalent in populations with diets high in animal protein and fat, while a *Prevotella*-dominated enterotype is more frequently found in individuals consuming high fiber [7]. This geographic and cultural variation in enterotype distribution highlights the significant role of the environment in shaping the gut microbiota. Previous studies have reported that the Korean population exhibits two distinct enterotypes, B-type and P-type, and aligns with our findings [28,29].

The analysis of gut microbiota by enterotype would allow for more precise assessments by grouping individuals with similar microbial profiles [30]. Previous studies have shown that enterotypes significantly influence the body’s responses to dietary components [12,31,32]. Given the distinct digestive functions and substrate preferences of *Bacteroides* and *Prevotella*, individuals with different enterotypes could react differently to specific diets [33]. In our study, we observed that garlic had a differential impact on gut microbiota composition based on enterotype. Specifically, individuals with *Bacteroides*-dominant microbiota responded more strongly to garlic treatment than those with *Prevotella*-dominant microbiota. This variation may be attributed to functional genes in the Bacteroides-dominant enterotype that facilitate the breakdown and utilization of specific garlic compounds. Taken together, our findings highlight the importance of personalized nutrition strategies based on microbial composition.

Since our study focused solely on the composition of the gut microbiome using 16S rRNA gene-based analysis, it does not provide detailed insights into the metabolic processes. To gain a deeper understanding, future studies should employ methods like shotgun metagenomic sequencing to identify the specific genes involved in garlic-treated metabolism. Unlike 16S rRNA sequencing, which only provides information about microbial composition, shotgun metagenomics allows for the comprehensive analysis of the entire genetic content of the microbial community [34,35]. This approach can identify functional genes and metabolic pathways that could be activated or suppressed in response to garlic consumption, such as those involved in sulfur metabolism, short-chain fatty acid production, and antimicrobial peptide synthesis. Additionally, untargeted metabolomics can profile metabolites produced by gut microbes, revealing the biochemical interactions between the host and the microbiome. Integrating metagenomics with metabolomics would provide a comprehensive view of the microbial ecosystem [36]. Combining these approaches would allow us to not only identify the genetic pathways affected by garlic but also to understand the resulting metabolic changes [37]. Our study was limited to microbial composition, but these findings suggest the importance of considering enterotype variations when examining dietary effects. This approach could enable the development of personalized nutritional strategies based on individual microbiome enterotypes. 

The fecal samples analyzed in this study were geographically confined to Korean participants. While we assessed the effects of garlic on gut microbiota by stratifying it into enterotypes, these findings should be further validated by studying participants from various geographical backgrounds. Given that additional enterotypes, such as the *Ruminococcus*-dominant type, have been reported, analyzing diverse populations is crucial for a more comprehensive understanding of garlic’s impact on the gut microbiome [9,10]. Furthermore, the relatively small sample size of the P-type group in this study may limit the generalizability of our results. Therefore, future research should include a broader range of geographic and ethnic groups, as well as larger sample sizes, to add to our findings.

The in vitro fecal incubation model used in this study does not fully capture the complexity of the human gut environment. This model lacks dynamic interactions that occur in vivo, such as host immune responses and hormonal influences. Moreover, the use of in vitro fecal incubation without simulated digestion may not fully mimic the complex processes of digestion and absorption that occur in the human body. In the in vitro model used in this study, garlic was provided directly to the gut microbiota in its undigested form, supplying nutrients and bioactive compounds directly to the microorganisms. This may lead to an over- or under-estimation of garlic’s impact on specific microbial populations and their functional activities. Therefore, while these in vitro findings offer valuable preliminary insights, they should be interpreted cautiously. However, our findings align with those of a study conducted in vivo, where garlic supplementation increased the abundance of *Bifidobacterium* in fecal samples, and suggests that our in vitro results may correspond with in vivo outcomes [38]. To address these limitations, future research could incorporate simulated digestion utilizing digestive enzymes. It would also be useful to conduct in vitro studies focusing on the undigested components of garlic, such as fructooligosaccharide, to better understand their specific effects on the gut microbiota [39]. Additionally, conducting human trials where the participants consume garlic and have their fecal samples analyzed would provide a more accurate reflection of garlic’s effects on the gut microbiota.

We confirmed that garlic promoted the growth of *B. adolescentis*, a crucial genus within the gut microbiome that plays a significant role in maintaining gut health. The *Bifidobacterium* can ferment dietary fibers into short-chain fatty acids (SCFAs) such as acetate, propionate, and butyrate [40]. These SCFAs not only provide energy for colonocytes but also support gut barrier integrity [41]. Beyond producing SCFAs, *Bifidobacterium* can synthesize essential nutrients like B vitamins, enhancing immune function and reducing inflammation [42]. These benefits suggest the crucial role of *Bifidobacterium* in maintaining gut health and reducing the risk of gastrointestinal disorders like irritable bowel syndrome (IBS) and inflammatory bowel disease (IBD) [43,44,45]. In this study, we found that garlic treatment could promote the growth of *B. adolescentis*, potentially acting as a prebiotic by selectively enhancing beneficial gut bacteria. This effect may contribute to improved gut health. However, further research is needed to determine whether these strains affected by garlic have a significant impact on the host’s health in vivo. Additionally, it is important to understand how garlic selectively enhances the growth of *B. adolescentis* in vitro among the different *Bifidobacterium* strains.

## 5. Conclusions

We highlighted the importance of considering enterotype variations when assessing the effects of plant-derived molecules like garlic on the gut microbiome. By grouping participants into distinct enterotypes, we demonstrated that the impact of diet could vary depending on the composition of the gut microbiome. These findings emphasize the potential for enterotype-based dietary strategies, suggesting that understanding an individual’s gut microbiome can lead to more personalized and effective nutritional interventions. Garlic is one of the gut-modulating agents that could alter the composition and functionality of the gut microbiome.

## Figures and Tables

**Figure 1 microorganisms-12-01971-f001:**
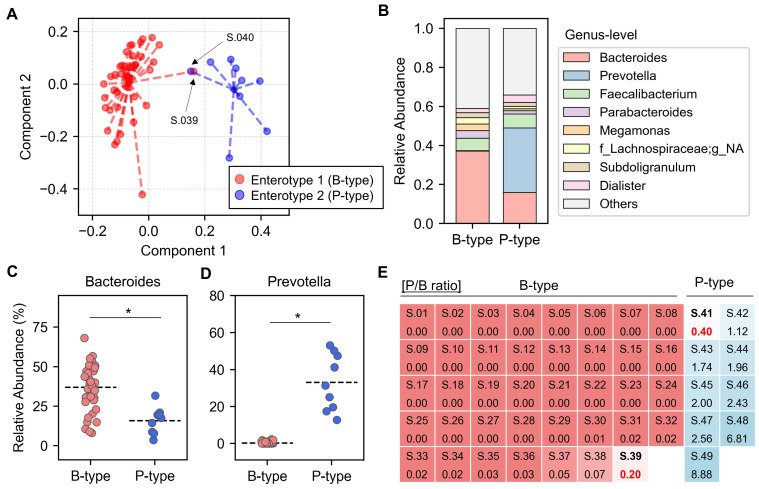
Gut enterotype classification. (**A**) Principal coordinates analysis was performed based on relative genus abundance utilizing the Jensen–Shannon divergence distance matrix and the partitioning around medoids clustering algorithm. Samples are represented as circles and distinguished by color, with Enterotype 1 (B-type) shown in red and Enterotype 2 (P-type) in blue. The dashed lines indicate the distance from the centroid (diamond) of each cluster. (**B**) The relative abundance of dominant bacterial genera is represented in B-type and P-type enterotypes. The g_NA represents genera not assigned at the genus level. The relative abundance of *Bacteroides* (**C**) and *Prevotella* (**D**) was compared within B-type and P-type enterotypes. Statistically significant differences (*p* < 0.05) are indicated with an asterisk (*). (**E**) A heatmap displays the *Prevotella* ratio [P/B ratio], which is calculated by dividing *Prevotella* levels by the combined levels of *Bacteroides* and *Prevotella*. Each sample is labeled with its name (S.xx) and specific P/B ratio. The color gradient ranges from red, indicating lower ratios, to blue, indicating higher ratios. Samples labeled in bold, S.39 and S.41, represent outliers with respect to the P/B ratio.

**Figure 2 microorganisms-12-01971-f002:**
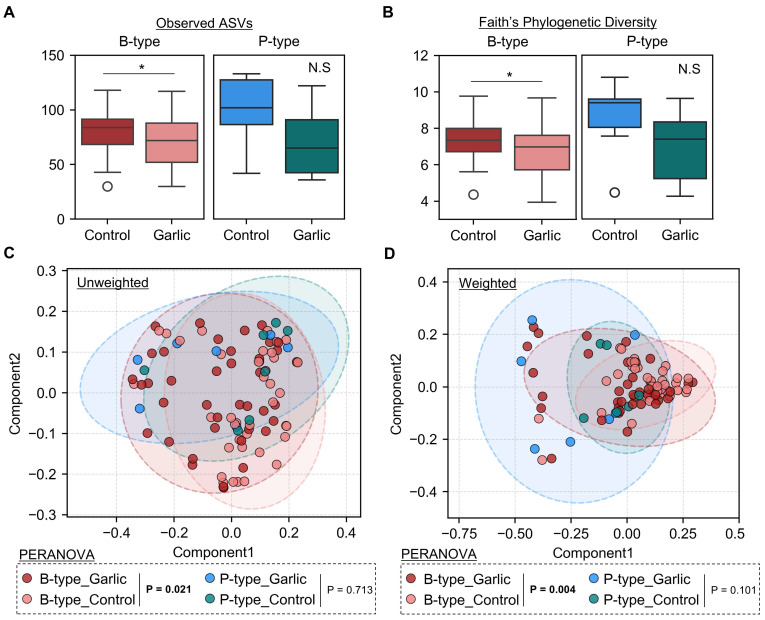
Effects of garlic treatment on microbial diversity. Alpha diversity was assessed using the number of observed amplicon sequence variants (**A**) and Faith’s phylogenetic diversity (**B**). Statistically significant differences are represented as: N.S., non-significant; *, *p* < 0.05. Microbial structure and composition of each group were analyzed through Principal Coordinates Analysis (PCoA), based on unweighted (**C**) and weighted (**D**) UniFrac distances. Each dot represents an individual sample. The ellipses represent 90% confidence intervals around the group centroids. The PERMANOVA results are presented with adjusted *p*-values (P) to indicate statistical significance.

**Figure 3 microorganisms-12-01971-f003:**
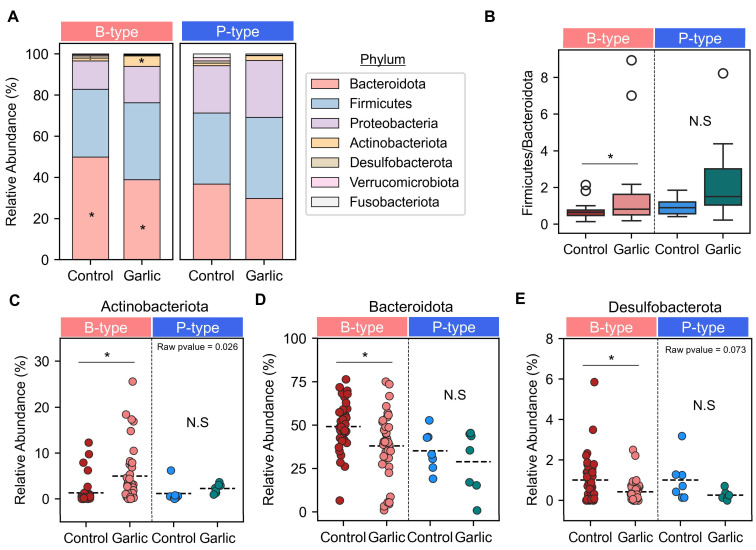
Phylum-level microbial change in response to garlic treatment. (**A**) The relative abundance of bacterial phyla in B-type and P-type enterotypes was assessed under control and garlic treatment conditions. (**B**) In both B-type and P-type enterotypes, the ratio of Firmicutes to Bacteroidota was compared between the control and garlic groups. The relative abundance of specific phyla, including Actinobacteriota (**C**), Bacteroidota (**D**), and Desulfobacterota (**E**), was compared within the enterotypes. Significant differences are marked as: *, adjusted *p*-value < 0.05; N.S., non-significant. Raw *p*-values are displayed over data. The colors represent different groups: red for the control group and pink for the garlic treatment group in the B-type; blue for the control group and green for the garlic treatment group in the P-type. Each circle represents an individual.

**Figure 4 microorganisms-12-01971-f004:**
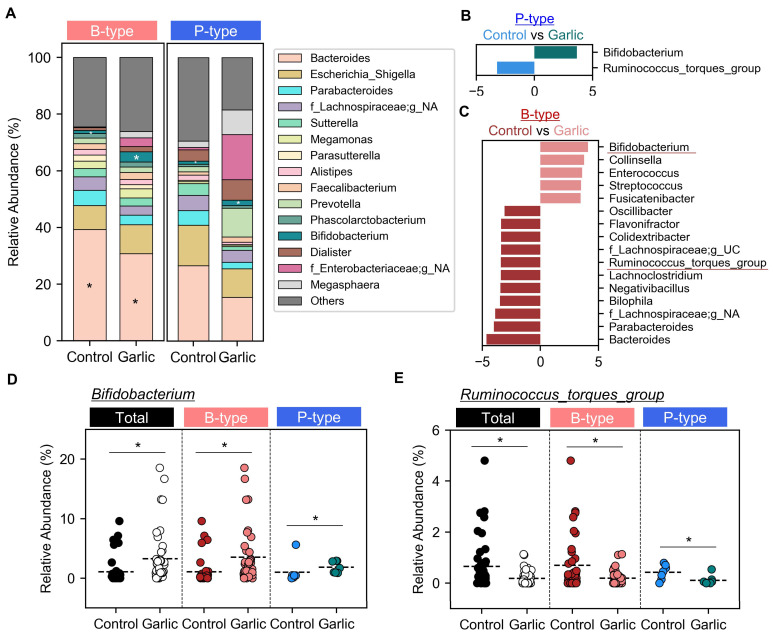
Effects of garlic treatment on microbial composition at the genus level. (**A**) The relative abundance of the top 15 bacterial genera in B-type and P-type enterotypes was assessed under control and garlic treatment conditions, with remaining genera grouped as others. The g_NA represents genera not assigned at the genus level. Differentially abundant taxa between the groups were identified using LEfSe in P-type (**B**) and B-type (**C**). The genera underlined in red represent the taxa that overlap in the P-type. The x-axis represents the LDA score. The relative abundance of *Bifidobacterium* (**D**) and *Ruminococcus*_torques_group (**E**) was compared across control and garlic treatment groups within B-type and P-type enterotypes. Significant differences are marked with an asterisk (*, *p* < 0.05). Each sample is shown as a circle. The colors indicate different groups: black for control group and white for garlic treatment group in the Total; red for control group and pink for garlic treatment group in the B-type; blue for control group and green for garlic treatment group in the P-type.

**Figure 5 microorganisms-12-01971-f005:**
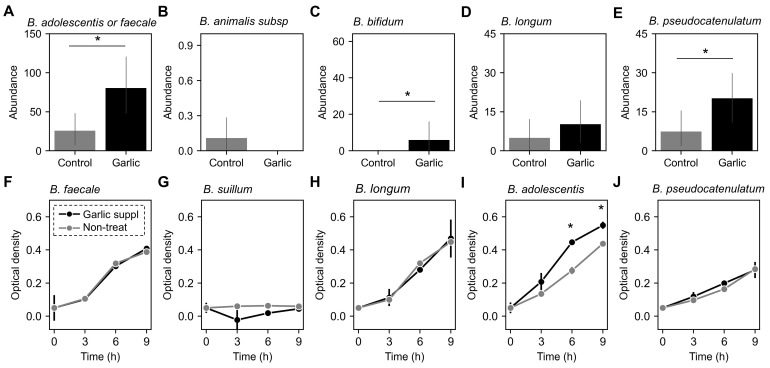
Prebiotic effects of garlic on specific bacterial taxa and growth. (**A**–**E**) The abundance of specific bacterial taxa at the ASV (amplicon sequence variant) level was assessed. These ASVs were mapped to the species level using BLAST analysis. The analyzed taxa include *Bifidobacterium adolescentis* or *faecale* (**A**), *Bifidobacterium animalis* subsp. (**B**), *Bifidobacterium bifidum* (**C**), *Bifidobacterium longum* (**D**), and *Bifidobacterium pseudocatenulatum* (**E**). (**F**–**J**) Growth curves of selected bacterial strains were measured in MRS medium with (Garlic suppl.) and without (Non-treat.) garlic supplementation. The optical density of *Bifidobacterium faecale* (**F**), *Bifidobacterium suillium* (**G**), *Bifidobacterium longum* (**H**), *Bifidobacterium adolescentis* (**I**), and *Bifidobacterium pseudocatenulatum* (**J**) was monitored over time. Black and gray represent the growth of the strain in MRS medium with and without garlic supplements, respectively. Statistically significant differences (*p* < 0.05) are indicated with an asterisk (*).

## Data Availability

All amplicon sequence data and metadata have been made public through the EMP data portal (Qiita, https://qiita.ucsd.edu; study ID: 15671) accessed on 5 September 2024.

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
