# Peer review of "Garlic-Induced Enhancement of Bifidobacterium: Enterotype-Specific Modulation of Gut Microbiota and Probiotic Populations"

_microorganisms, 2024, doi:10.3390/microorganisms12101971_

Round 1

Reviewer 1 Report

Comments and Suggestions for Authors

The authors investigated the effect of garlic on the human gut microbiome using an in vitro fecal incubation model based on fecal enterotype, and whether garlic had prebiotic potential by stimulating the growth of Bifidobacterium species. Of the 48 individuals analyzed, 40 had a Bacteroides-dominant enterotype (B-type), whereas eight had a Prevotella-dominant enterotype (P-type). Garlic treatment significantly reduced the microbial richness of B-type individuals, as well as phylogenetic diversity. On the other hand, in P-type individuals, alpha diversity was not significantly affected by garlic treatment. The same was observed for beta diversity. At the genus level, the analyses showed that individuals with the P-type enterotype presented significant changes in relative abundance only with bacteria of the genus Bifidobacterium (increase) and Ruminococcus (decrease), while those with the type presented changes in 16 genera, including these. The analysis of the prebiotic effect in vitro showed a stimulation of growth only for B. adolescentis, whereas the analysis of relative abundance showed significant differences for three species of Bifidobacterium, including B. adolescentis. The authors concluded that garlic could affect the composition of the intestinal microbiome, showing prebiotic potential. The results of this study are interesting and the manuscript is acceptable for publication. However, some modifications are required before it is accepted. Here are some comments:

1) The number of individuals with type B enterotypes was much higher than that with type P. Could this not have affected the results? I suggest that a comment be included on this;

2) An important limitation of the study is that the results were based on an in vitro incubation model. Therefore, compounds with the potential to modulate the microbiome were not subjected to digestive enzymes or acidic pH. Thus, the authors could include in the discussion section studies that described the presence of known prebiotics in garlic, which could reiterate their findings. Some compounds, such as FOS and inulin, are recognized as prebiotics that resist digestion and reach the large intestine without alterations. Please include articles describing the prebiotic compounds in garlic ( Rahim et al. Optimization of the ultrasound operating conditions for extraction and quantification of Fructooligosaccharides from garlic (Allium sativum L.) via high-performance liquid chromatography with refractive index detector. Molecules, v. 27, n. 19, p. 6388, 2022.);

3) Was the in vitro study with Bifidobacterium species performed with reconstituted MRS, but without glucose, or was it performed with a complete culture medium (with glucose)? This assay should have been performed in the absence of glucose, as the presence of two carbon sources would not allow a more adequate evaluation of the prebiotic potential of garlic.

Minor comments:

Line 200 and 202: Actinobacteriota is misspelled;

Author Response

# Reviewer1

The authors investigated the effect of garlic on the human gut microbiome using an in vitro fecal incubation model based on fecal enterotype, and whether garlic had prebiotic potential by stimulating the growth of Bifidobacterium species. Of the 48 individuals analyzed, 40 had a Bacteroides-dominant enterotype (B-type), whereas eight had a Prevotella-dominant enterotype (P-type). Garlic treatment significantly reduced the microbial richness of B-type individuals, as well as phylogenetic diversity. On the other hand, in P-type individuals, alpha diversity was not significantly affected by garlic treatment. The same was observed for beta diversity. At the genus level, the analyses showed that individuals with the P-type enterotype presented significant changes in relative abundance only with bacteria of the genus Bifidobacterium (increase) and Ruminococcus (decrease), while those with the type presented changes in 16 genera, including these. The analysis of the prebiotic effect in vitro showed a stimulation of growth only for B. adolescentis, whereas the analysis of relative abundance showed significant differences for three species of Bifidobacterium, including B. adolescentis. The authors concluded that garlic could affect the composition of the intestinal microbiome, showing prebiotic potential. The results of this study are interesting and the manuscript is acceptable for publication. However, some modifications are required before it is accepted. Here are some comments:

Q1) The number of individuals with type B enterotypes was much higher than that with type P. Could this not have affected the results? I suggest that a comment be included on this;

Answer: Thanks for this comment. We acknowledge the reviewer's concerns regarding the limited number of participants in the Prevotella-dominant (P-type) group, which could potentially limit the generalizability of our findings. To address this issue, we have discussed the necessity for future studies to incorporate larger sample sizes specifically for the P-type enterotype.

(Lines 326-339) “Furthermore, the relatively small sample size of the P-type group in this study may limit the generalizability of our results. Therefore, future research should include a broader range of geographic and ethnic groups, as well as larger sample sizes, to validate our findings.”

Q2) An important limitation of the study is that the results were based on an in vitro incubation model. Therefore, compounds with the potential to modulate the microbiome were not subjected to digestive enzymes or acidic pH. Thus, the authors could include in the discussion section studies that described the presence of known prebiotics in garlic, which could reiterate their findings. Some compounds, such as FOS and inulin, are recognized as prebiotics that resist digestion and reach the large intestine without alterations. Please include articles describing the prebiotic compounds in garlic ( Rahim et al. Optimization of the ultrasound operating conditions for extraction and quantification of Fructooligosaccharides from garlic (Allium sativum L.) via high-performance liquid chromatography with refractive index detector. Molecules, v. 27, n. 19, p. 6388, 2022.);

Answer: Thanks for this comment. As the reviewer pointed out, careful interpretation of the results obtained using the in vitro fecal incubation model is crucial. In this study, to strengthen the reliability of our results, we noted that our findings align with those from in vivo studies (lines 333-336). Furthermore, we have incorporated the recommended citation and discussed the potential impact of undigested garlic components on the gut microbiome, as bellow.

(Lines 342-345) “To address these limitations, future research could incorporate simulated digestion using digestive enzymes. It would also be useful to conduct in vitro studies focusing on undigested components of garlic, such as fructooligosaccharide, to better understand their specific effects on the gut microbiota (Rahim et al. 2022).”

Q3) Was the in vitro study with Bifidobacterium species performed with reconstituted MRS, but without glucose, or was it performed with a complete culture medium (with glucose)? This assay should have been performed in the absence of glucose, as the presence of two carbon sources would not allow a more adequate evaluation of the prebiotic potential of garlic.

Answer: Thank you for pointing out the need for more detailed descriptions of our experimental conditions. We have updated the “Materials and Methods” section of our manuscript to include the following clarification:

(Lines 107-109) “The strains were cultured anaerobically at 37°C in MRS broth, which was supplemented with 0.1% (w/v) L-cysteine but excluded glucose.”

Minor comments:

Line 200 and 202: Actinobacteriota is misspelled;

Answer: Thanks. We have corrected "Actinobacteria" to "Actinobacteriota."

Reviewer 2 Report

Comments and Suggestions for Authors

The article “Garlic-Induced Enhancement of Bifidobacterium: Enterotype Specific Modulation of Gut Microbiota and Probiotic Populations” studies the effects of garlic on the gut microbiome using an in vitro human fecal incubation model. For this purpose, bacteroides-dominant enterotype was evaluated showing significant changes in overall microbial diversity in response to garlic, while the Prevotella dominant enterotype was not affected. In general, the document is properly written resulting in relevance to the scientific community. The introduction describes the composition and importance of the gut microbiome. The third paragraph describes the properties of garlic as a medicinal plant. The aim of the study is given in the final paragraph. A brief and concise Materials and Methods section mentions the procedures and statistical analysis performed on the data. The results are supported with nice figures showing the statistical outcomes. The discussion section is properly written. To improve the quality of the document I suggest to:

1.- Introduction section, line 65 stated that the specific effects of garlic on the human microbiome remain poorly understood. I consider there is a lack of connection and justification between the gut microbiome and the garlic properties described in the first and second paragraphs of the introduction section. It is necessary to include a more solid argument for the study.

2.- Subsection 2.4 Bacterial growth rate, states in line 104 that bacterial strains were grown anaerobically at 37°C in MRS broth. It is necessary to name the bacteria grown in the study. In which part of the Methods are mentioned the Bacteroides and the Prevotella

3.- I suggest including the text of lines 326 to 333 in a conclusion section.

Author Response

# Reviewer2

The article “Garlic-Induced Enhancement of Bifidobacterium: Enterotype Specific Modulation of Gut Microbiota and Probiotic Populations” studies the effects of garlic on the gut microbiome using an in vitro human fecal incubation model. For this purpose, bacteroides-dominant enterotype was evaluated showing significant changes in overall microbial diversity in response to garlic, while the Prevotella dominant enterotype was not affected. In general, the document is properly written resulting in relevance to the scientific community. The introduction describes the composition and importance of the gut microbiome. The third paragraph describes the properties of garlic as a medicinal plant. The aim of the study is given in the final paragraph. A brief and concise Materials and Methods section mentions the procedures and statistical analysis performed on the data. The results are supported with nice figures showing the statistical outcomes. The discussion section is properly written. To improve the quality of the document I suggest to:

Q1.- Introduction section, line 65 stated that the specific effects of garlic on the human microbiome remain poorly understood. I consider there is a lack of connection and justification between the gut microbiome and the garlic properties described in the first and second paragraphs of the introduction section. It is necessary to include a more solid argument for the study.

Answer: Thanks for this comment. We have revised the introduction in accordance with the reviewer’s suggestions. We have provided a more detailed discussion (description) on the importance of studying the effects of garlic on the human gut microbiome.

(Lines 64-68) “While the beneficial effects of garlic on human health are well-known, its role in shaping the composition and function of the gut microbiome is not fully understood. There is limited insight into how garlic may serve as a prebiotic, selectively promoting the growth of beneficial microbial populations and, consequently, influencing overall human health.”

Q2.- Subsection 2.4 Bacterial growth rate, states in line 104 that bacterial strains were grown anaerobically at 37°C in MRS broth. It is necessary to name the bacteria grown in the study. In which part of the Methods are mentioned the Bacteroides and the Prevotella

Answer: Thanks. We have included detailed information about the bacterial strains used in this study in Supplementary Table S1. Moreover, the results for Bacteroides and Prevotella were derived from the in silico analysis described in section 2.3 (16rRNA amplicon analysis). Since these genera were not part of the in vitro culturing, they are not included in the supplementary table.

(Line 107) “The bacterial strains used in this study were listed in the Table S1.”

(In Supplementary data)

Table S1. The bacterial strains used in this study

Strains

Origin

Source

Bifidobacterium adolescentis

Human feces of a healthy, 27-year-old male, Korea

In this lab

Bifidobacterium faecale

Human feces of a two-week-old baby, Korea

In this lab

Bifidobacterium longum

Human feces

In this lab

Bifidobacterium suillum

Feces of piglets, Bologna, Italy

In this lab

Bifidobacterium pseudocatenulatum

Human feces, Korea

In this lab

Q3.- I suggest including the text of lines 326 to 333 in a conclusion section.

Answer: Thanks for this comment. As the reviewer’s suggestion, we incorporated this paragraph into the conclusion.

(Lines 363-371) 5. Conclusions

“We highlight the importance of considering enterotype variations when assessing the effects of plant-derived molecules like garlic on the gut microbiome. By grouping participants into distinct enterotypes, we demonstrated that the impact of diet could vary de-pending on the composition of the gut microbiome. These findings emphasize the potential for enterotype-based dietary strategies, suggesting that understanding an individual's gut microbiome can lead to more personalized and effective nutritional interventions. Garlic is one of the gut modulating agents that could alter the composition and functionality of the gut microbiome.”

Reviewer 3 Report

Comments and Suggestions for Authors

This manuscript investigated how garlic affects the gut microbiome, which is a timely and relevant subject in microorganisms. The approach of stratifying gut microbiota into B-type and P-type allows for a more nuanced understanding of effects of garlic on different gut microbial compositions, which is a novel methodological choice. The analysis on the prebiotic potential of garlic in promoting the growth of specific Bifidobacterium species adds a practical dimension to the research, linking findings to potential dietary interventions. However, there are some details that need to be further refined before publication. 

Line 130-131: the relatively small sample size, especially for the P-type enterotype, may limit the generalizability of the results.

Line 214-216: the study focuses heavily on microbiome composition without delving deeply into functional or metabolic outcomes of garlic consumption. Can we consider adding functional pathway analysis of microorganisms or adding relevant content in the discussion?

Line 263-274: Consider discussing more thoroughly how geographic and dietary factors could influence enterotype distribution, especially since the participants are all Korean.

Line 283: The authors briefly mention the potential of shotgun metagenomics but do not explore this in-depth.

Line 315-325: The use of an in vitro fecal incubation model is practical but inherently limited in replicating the complexities of human digestion and gut interaction. The study acknowledges this, but a more detailed discussion of how this limitation could impact the applicability of the results to in vivo conditions is warranted.

The manuscript could benefit from a deeper exploration of why garlic has such a pronounced effect on the B-type but not the P-type enterotype. A more detailed hypothesis regarding the molecular mechanisms involved in these differences would be valuable.

The findings for the P-type are relatively underdeveloped compared to the B-type. Expanding on the possible reasons behind the limited effect of garlic in this group would provide a more balanced view.

Author Response

# Reviewer3

This manuscript investigated how garlic affects the gut microbiome, which is a timely and relevant subject in microorganisms. The approach of stratifying gut microbiota into B-type and P-type allows for a more nuanced understanding of effects of garlic on different gut microbial compositions, which is a novel methodological choice. The analysis on the prebiotic potential of garlic in promoting the growth of specific Bifidobacterium species adds a practical dimension to the research, linking findings to potential dietary interventions. However, there are some details that need to be further refined before publication.

Q1. Line 130-131: the relatively small sample size, especially for the P-type enterotype, may limit the generalizability of the results.

Answer: Thanks for this comment. We agree the reviewer’s concern about the small sample size, especially for the P-type enterotype, which may limit the generalizability of our findings on the effects of garlic on the gut microbiome. In the revised manuscript, we have addressed this limitation and emphasized the importance of future studies with larger and more balanced sample sizes to validate these results.

(Lines 326-329) “Furthermore, the relatively small sample size of the P-type group in this study may limit the generalizability of our results. Therefore, future research should include a broader range of geographic and ethnic groups, as well as larger sample sizes, to validate our findings.”

Q2. Line 214-216: the study focuses heavily on microbiome composition without delving deeply into functional or metabolic outcomes of garlic consumption. Can we consider adding functional pathway analysis of microorganisms or adding relevant content in the discussion?

Answer: Thanks for this comment. As our study focused on microbiome composition using 16S rRNA sequencing, we are unable to accurately assess functional or metabolic outcomes. As noted in the limitations of our study, a shotgun metagenomics and metabolomics approach would be necessary. Such methods would enable a more precise analysis of functional pathways and provide deeper insights into the metabolic effects of garlic consumption. We have addressed this limitation in the discussion and emphasized the need for future studies to incorporate shotgun metagenomic analysis to understand the functional impacts of garlic on the microbiome.

(Line 307-317) “Unlike 16S rRNA sequencing, which only provides information about microbial composition, shotgun metagenomics allows for the comprehensive analysis of the entire genetic content of the microbial community (Li et al. 2023; Quince et al. 2017). This approach can identify functional genes and metabolic pathways that could be activated or suppressed in response to garlic consumption, such as those involved in sulfur metabolism, short-chain fatty acid production, and antimicrobial peptide synthesis. Additionally, untargeted metabolomics can profile metabolites produced by gut microbes, revealing the biochemical interactions between the host and the microbiome. Integrating metagenomics with metabolomics would provide a comprehensive view of the microbial ecosystem (Vernocchi et al. 2016). Combining these approaches allows us to not only identify the genetic pathways affected by garlic but also to understand the resulting metabolic changes (Bai et al. 2024).”

Q3. Line 263-274: Consider discussing more thoroughly how geographic and dietary factors could influence enterotype distribution, especially since the participants are all Korean.

Answer: As suggested by the reviewer, we have added a discussion on how geographic and dietary factors could influence enterotype distribution.

(Line 279-290) “Enterotypes, the classification of human gut microbiota into distinct compositional profiles, could provide significant insights into the relationship between microbiota composition and host factors. The three primary enterotypes are Bacteroides-dominant type (enterotype 1), Prevotella-dominant type (enterotype 2), and Ruminococcus-dominant type (enterotype 3) (Arumugam et al. 2011, Qin et al. 2012). These enterotypes are not randomly distributed but are profoundly influenced by dietary patterns, which reflect geographic and cultural differences. For instance, a Bacteroides-dominated enterotype is prevalent in populations with diets high in animal protein and fat, while a Prevotella-dominated enterotype is more frequently found in individuals consuming high-fiber (Wu et al. 2011). This geographic and cultural variation in enterotype distribution highlights the significant role of the environment in shaping the gut microbiota. Previous studies have reported that the Korean population exhibits two distinct enterotypes, B-type and P-type, aligning with our findings (Lim et al. 2014, Lim et al. 2021).”

Q4. Line 283: The authors briefly mention the potential of shotgun metagenomics but do not explore this in-depth.

Answer: Thanks. We have expanded the discussion to highlight the advantages of incorporating shotgun metagenomics in future research, as this would allow for a more detailed exploration of functional pathways and deeper insights into the effects of garlic on the gut microbiome. We also emphasize the importance of metabolomics studies to complement these findings. A multi-omics approach, integrating metagenomics and metabolomics, would provide a comprehensive understanding of how garlic influences the microbiome at the molecular level, enhancing our knowledge of its impact on microbial function and host health.

(Line 307-317) “Unlike 16S rRNA sequencing, which only provides information about microbial composition, shotgun metagenomics allows for the comprehensive analysis of the entire genetic content of the microbial community (Li et al. 2023; Quince et al. 2017). This approach can identify functional genes and metabolic pathways that could be activated or suppressed in response to garlic consumption, such as those involved in sulfur metabolism, short-chain fatty acid production, and antimicrobial peptide synthesis. Additionally, untargeted metabolomics can profile metabolites produced by gut microbes, revealing the biochemical interactions between the host and the microbiome. Integrating metagenomics with metabolomics would provide a comprehensive view of the microbial ecosystem (Vernocchi et al. 2016). Combining these approaches allows us to not only identify the genetic pathways affected by garlic but also to understand the resulting metabolic changes (Bai et al. 2024).”

Q5. Line 315-325: The use of an in vitro fecal incubation model is practical but inherently limited in replicating the complexities of human digestion and gut interaction. The study acknowledges this, but a more detailed discussion of how this limitation could impact the applicability of the results to in vivo conditions is warranted.

Answer: We recognize the limitations of using an in vitro fecal incubation model, as it does not fully replicate the complexities of human digestion and gut interactions. In this model, garlic was provided directly to the gut microbiota in its undigested form, which may lead to an over- or underestimation of its impact on microbial populations. To strengthen our results, we noted that our findings align with those from in vivo studies, thereby supporting the reliability of our results. We have addressed the reviewer’s concerns in the discussion, emphasizing the need for more advanced models, such as simulated digestion systems or human trials, to accurately elucidate the effects of garlic, as bellow.

(Lines 334-339) “In the in vitro model used in this study, garlic was provided directly to the gut microbiota in its undigested form, supplying nutrients and bioactive compounds directly to the microorganisms. This may lead to an over- or underestimation of garlic’s impact on specific microbial populations and their functional activities. Therefore, while these in vitro findings offer valuable preliminary insights, they should be interpreted cautiously.”

Q6. The manuscript could benefit from a deeper exploration of why garlic has such a pronounced effect on the B-type but not the P-type enterotype. A more detailed hypothesis regarding the molecular mechanisms involved in these differences would be valuable. The findings for the P-type are relatively underdeveloped compared to the B-type. Expanding on the possible reasons behind the limited effect of garlic in this group would provide a more balanced view.

Answer: Thanks. We agree with the reviewer that our study cannot precisely explain why garlic has a pronounced effect on the B-type enterotype but not on the P-type. The current experimental data do not provide enough detail to support a molecular-level explanation for these differences. We infer that the differing responses may be attributed to variations in microbial composition and the genetic capacity to metabolize garlic compounds between the two enterotypes. However, we lack sufficient evidence to confirm this hypothesis. Therefore, we have briefly mentioned it as a potential explanation and emphasized the need for further research to explore the underlying molecular mechanisms. Our study does not fully elucidate why garlic affects certain enterotypes more than others, as we lack the data for a more detailed explanation. Therefore, we suggest that further research on this issue be included in the revised manuscript. However, our findings highlight the importance of considering enterotype variations when examining dietary effects. Understanding these variations could pave the way for developing personalized nutritional strategies.

(Line 292-302) “Previous studies have shown that enterotypes significantly influence the body's responses to dietary components (Christensen et al. 2018; Choi et al. 2021; Fu et al. 2022). Given the distinct digestive functions and substrate preferences of Bacteroides and Prevotella, individuals with different enterotypes could react differently to specific diets (Chen et al. 2017). In our study, we observed that garlic had a differential impact on gut microbiota composition based on enterotype. Specifically, individuals with Bacteroides-dominant microbiota responded more strongly to garlic treatment than those with Prevotella-dominant microbiota. This variation may be attributed to functional genes in the Bacteroides-dominant enterotype that facilitate the breakdown and utilization of specific garlic compounds. Taken together, our findings highlight the importance of personalized nutrition strategies based on microbial composition.”